# Inhibitory Activities of Dimeric Ellagitannins Isolated from *C**ornus alba* on Benign Prostatic Hypertrophy

**DOI:** 10.3390/molecules26113446

**Published:** 2021-06-06

**Authors:** Dong-Hui Park, Kwan-Hee Park, Jun Yin, Min-Ji Kim, Seong-Eun Yoon, Sun-Ho Lee, Jun-Hyeok Heo, Hyun-Joo Chung, Jin-Wook Kim, Kyung-Mi Kim, Min-Won Lee

**Affiliations:** 1Laboratory of Pharmacognosy and Natural Product Derived Medicine, College of Pharmacy, Chung-Ang University, Seoul 06974, Korea; donghee9611@naver.com (D.-H.P.); kwany1982@naver.com (K.-H.P.); yinjun89@naver.com (J.Y.); kam4256@naver.com (M.-J.K.); blue6462@naver.com (S.-E.Y.); truman2001@naver.com (S.-H.L.); fifasthur@naver.com (J.-H.H.); 2Department of Urology, College of Medicine, Chung-Ang University, Seoul 06974, Korea; hyunjoo1121@cau.ac.kr (H.-J.C.); jinwook@cau.ac.kr (J.-W.K.); 3Life Science Research Institute, NOVAREX.CO., Ltd., Cheongju 363885, Korea; kkm3507@novarex.co.kr

**Keywords:** dimeric ellagitannin, *Cornus alba*, benign prostatic hypertrophy (BPH)

## Abstract

Benign prostatic hypertrophy (BPH) is an intractable chronic inflammatory disease. We studied the efficacy of two ellagitannins, namely camptothin B (**1**) and cornusiin A (**2**) that were isolated from *Cornus alba* (CA) for the treatment of BPH, which is a common health issue in older men. The ellagitannins (**1** and **2**) were evaluated on its inhibitory activities of the enzyme 5α-reductase and tumor necrosis factor (TNF)-α, its interleukin (IL)-1β, IL-6, and IL-8 production, and its anti-proliferation and apoptosis induction in prostate cells that show hypertrophy (RWPE-1 cell). In inhibition of 5α-reductase, the ellagitannins (**1** and **2**) showed potential effects, compared to the positive control, finasteride. In the case of IL-1β, IL-6, IL-8, and TNF-α, **1** and **2** showed good inhibitory effects as compared to the control group treated with LPS. The ellagitannins (**1** and **2**) were also shown to inhibit proliferation of, and induce apoptosis in, the RWPE-1 cell. These results suggest that the ellagitannins (**1** and **2**) may be good candidates for the treatment of BPH.

## 1. Introduction

Benign prostatic hypertrophy (BPH), a type of overgrowth of the prostate tissue, is a common problem in older men. Prostate hyperplasia is closely related to aging and male hormones, and is a progressive disease that does not directly affect life with increase in age, but lowers urinary symptoms, such as residual, empty, thin, night, and intermittent urination along with mental, social, sexual problems [1,2,3]. Studies have reported that prostatic growth begins at age 30, with BPH affecting approximately 50% of men aged 50 and 70% of men older than 80 [4,5]. The cause of BPH is unclear, however, studies have suggested that the common factors that might induce the development of BPH are hormonal alterations, ageing, and inflammation [6,7]. The main treatments for BPH are medications and surgery. However, few men have symptoms or other problems that are severe enough to require surgical treatment. The medications used to treat BPH include alpha-blockers and 5-alpha reductase inhibitors (5-ARI) [8]. Alpha blockers treat the symptoms of BPH by relaxing the smooth muscle tissue in the prostate and bladder to help the urine flow out. It has no effect on prostate growth through its mechanism of action, and can quickly heal symptoms of the lower urinary tract [9]. BPH is caused by the influence of dihydrotestosterone (DHT), which is known to be a testosterone derivative with a high affinity for androgen receptors [8]. In the case of 5-ARI, the size of the prostate is reduced by inhibiting the production of DHT. Unlike the alpha blocker, which improves relatively quickly, 5-ARI takes a long time to contract the prostate [10].

*Cornus alba* (CA), is a species of Cornaceae, native to Siberia, Northern China, and Korea. It is a deciduous shrub that can grow to 3 m high, like a small tree. The barks and leaves of CA were used as folk herbal medicine for the treatment of inflammation and hemolysis. CA bark is also used in hemoptysis of pulmonary tuberculosis, and the root is used for fever and cold [11,12]. *Cornus* species are known to have antioxidant, anti-inflammatory, and anti-cancer biological activities [13,14].

In our previous work, we studied the antiproliferative effects of new dimeric ellagitannin from *Cornus alba* in prostate cancer cells, including apoptosis-related S-phase arrest [15]. This study was performed to prove that ellagitannins (**1** and **2**), which were isolated from CA, can be used as treatment for BPH, through the evaluations of cytokines IL-6, IL-8, IL-1β, TNF-α, anti-proliferative effect on BPH cell line, flow cytometry analysis of apoptosis, and 5α-reductase inhibition activity.

## 2. Result

### 2.1. Phytochemicals from Cornus alba

We isolated several hydrolysable tannins from CA and selected two major ellagitannins (**1** and **2**) for this study (Figure 1) [16].

### 2.2. Inhibition of Cytokines Production

Compounds **1** and **2** were shown to potently inhibit the production of cytokines IL-6 and 8, which are autocrine growth factors that play a key role in inducing the development of hyperplasia of the prostate in BPH patients and in prostate cancer cells (Figure 2 and Figure 3) [17,18].

The ellagitannins (**1** and **2**) were also shown to induce a potent decrease in the productions of IL-1β and TNF-α (Figure 4 and Figure 5), which are major cytokines involved in systemic inflammation, as well as in stimulating the acute phase reaction and biological markers of prostatic secretion, as indicators of prostatic inflammation in chronic prostatitis [19,20,21].

### 2.3. Anti-Proliferative Effect on BPH Cell Line

The anti-proliferative effect of the compounds on the BPH cell line (RWPE-1 cell) was measured via an MTT assay, after the cells were treated with the compounds at concentrations between 6.25–100 μM, for 24 h. The ellagitannins (**1** and **2**) were shown to have anti-proliferative activities with half maximal inhibitory concentrations (IC_50_) of 3.38 ± 0.71 and 1.54 ± 0.21, respectively.

### 2.4. Flow Cytometry Analysis of Apoptosis

Phosphatidylserine (PS) is a phospholipid that exists inside the cell membrane in normal cells, and is exposed to the outside of the cell membrane, when apoptosis progresses [22]. PS binds to a specific protein called Annexin V. By combining a fluorescent substance here, the degree of apoptosis of cells can be checked through flow cytometry [23]. In addition, using a propidium iodide (PI) fluorescent dye that passes through the cell membrane of dead cells and binds to DNA, it is possible to identify early apoptosis, late apoptosis, and even necrosis. In Figure 6 and Figure 7, the uniform viability, which shows the lower left part, represents a living normal cell, and both Annexin V and PI are observed as negative. The ellagitannins (**1** and **2**) represent 15.14% and 8.39%, respectively. Early apoptosis, indicating the lower right part, was found to be positive only for Annexin V, with 6.06% and 7.95% of **1** and **2**, respectively. Late apoptosis, which indicates the upper right part, indicates cells that have died due to apoptosis progression. Additionally, because the cell membrane collapses, both Annexin V and PI are positive and the ellagitannins (**1** and **2**) were 43.00% and 74.43%, respectively. Necrosis, which represents the upper left part, is a cell necrosis or pathologically dead part, PI is positive because PS is not exposed to the extracellular membrane, but Annexin V is negative and the ellagitannins (**1** and **2**) represent 35.79% and 9.24%, respectively.

Cells were incubated with the compounds **1** and **2** at a concentration of 25 μM for 24 h, in a complete medium. The percentage of cell in each quadrant is indicated (lower left—normal, lower right—apoptosis, upper right—late apoptosis, and upper left—necrosis)

The percentage of cells in the upper left (necrotic cells) and the upper right (late apoptotic cells) quadrants. The percentage of cells in the lower right (early apoptotic cells), and lower left (viable cells) portions of the histogram was calculated for comparison.

### 2.5. 5α-Reductase Inhibition Activity

The reduction of the level of DHT via the inhibition of 5α-reductase might be a good way to treat BPH [24]. As such, 5α-reductase and testosterone were mixed with the samples and the concentration of testosterone was detected by HPLC. Compared to the samples treated with the control (finasteride), the ellagitannins (**1** and **2**) showed potential 5 α-reductase inhibitory activity (Figure 8).

## 3. Discussion

There is ongoing research that aims to find a therapeutic agent for prostate disease using natural products through evaluation of the pharmacological properties of anti-proliferative, antioxidant, and anti-inflammatory substances [25].

In this study, the ellagitannins (**1** and **2**) isolated from CA were evaluated for their inhibitory activities on cytokines IL-6, IL-8, IL-1β, and TNF-α; their anti-proliferative effect on the RWPE-1 cell; flow cytometry analysis of apoptosis; and 5α-reductase inhibition activity.

The overexpression of proinflammatory cytokines, such as interferon (IFN)-γ and IL-17, in the BPH tissue, induces the production of IL-6 and IL-8 [20]. It is reported that IL-1β and tumor necrosis factor (TNF)- α were shown to be elevated in BPH patients [16].

The ellagitannins (**1** and **2**) potently reduced the productions of the cytokines, IL-6 and IL-8 (Figure 2 and Figure 3), and IL-1β and TNF-α (Figure 4 and Figure 5).

The ellagitannins (**1** and **2**) were shown to have anti-proliferative activities on the BPH cell line (RWPE-1 cell, IC50 of 3.38 ± 0.71 and 1.54 ± 0.21, respectively).

Their effect on apoptosis were then further analyzed using an Annexin V-FITC (Annexin V) and propidium iodide (PI) staining assay. Annexin V, a Ca^2+^-dependent phospholipid (PS)-binding protein, binds to the exposed PS at the surface of the apoptotic cell membrane [26,27], while PI, a DNA staining agent, cannot cross the intact plasma membrane of live cells [28]. The Annexin V+/PI− cells were considered to be early apoptotic cells, while the Annexin V+/PI+ cells were considered to be late apoptotic cells. The results indicated that **1** and **2** exhibited potent late apoptosis activity. In addition, the ratio of necrosis was lower than that of **2** to **1**, and the overall apoptosis rate was higher, so it could be said that **2** was more effective than **1** (Figure 6 and Figure 7).

The enzyme 5α-reductase catalyzed the reduction of testosterone to dihydrotestosterone (DHT), which could bind to the androgen receptor (AR). The AR, which is activated by binding to the androgenic hormones, testosterone, and DHT, regulates gene expression via the DNA-binding transcription factors [29]. Moreover, DHT has a stronger affinity for human AR than testosterone and adrenal androgens [30,31]. The results revealed that the ellagitannins (**1** and **2**) have potent inhibition of 5α-reductase.

## 4. Materials and Methods

### 4.1. Compounds

CA (5.7 kg) was collected at the Korea National Arboretum (Pocheon, Korea). The CA samples (5.7 kg) were pulverized and extracted with 80% acetone at room temperature to obtain the CA extract (463 g). Then, the extraction and its subfraction (356 g) were separated via repeated column chromatography. Next, the samples were dissolved in water and filtered through a Celite 545 (Duksan Pure Chemical, Ansan, Korea) filter. After this, 356 g of a water-soluble fraction was obtained together with 89 g of water-insoluble residue. Only 243 g of the water-soluble fraction was loaded onto a Sephadex LH-20 column (15 × 100 cm), equilibrated with water. After the column was eluted with a water–methanol gradient system and washed with 60% acetone, 14 fractions were obtained. Fraction 13 (31.6 g) was loaded onto an MCI CHP 20P column (5 × 60 cm), with a water–methanol gradient system, and five subfractions were obtained. Afterwards, fraction 13-2 was loaded onto a Daisogel column (3 × 50 cm), with a water 20% methanol gradient in a medium pressure liquid Chromatography (MPLC) system (5 mL/min, 280 nm). It was further separated by column chromatography on the Sephadex LH-20 column (10 × 80 cm), using a water–methanol 60% acetone gradient system. As a result, cornusiin A (2, 1.2 g) was obtained. Fraction 13-3 was loaded onto a Sephadex LH-20 column (2.5 × 50 cm), with a water–methanol 60% acetone gradient system, and was further separated by column chromatography on the MCI CHP20P column (5 × 60 cm), using a water–methanol gradient system. This resulted in three additional sub-fractions. Fraction 13-3-1 was then separated by column chromatography using a Toyopearl HW-40 column (2.5 × 50 cm), with a 70% methanol−70% acetone (10:0→7:3) gradient, which yielded camptothin B (1, 815 mg).

#### 4.1.1. Camptothin B (1) 

The product was an amorphous brown powder. Structural data were as follows: 

LRFAB-MS *m*/*z*: 1721 [M − H]^(−)^; CD (MeOH): [θ]_223_ 16.13, [θ]_259_ 3.18, [θ]_280_ 6.36; 

^1^H-NMR (600 MHz, Acetone-*d*_6_ + D_2_O): *δ* 3.73–3.96 (each br d, *J* = 13.2, H-6b of each two form), 4.14, 4.48, 4.59–4.65 (br dd, *J* = 6.0, 9.6 Hz, H-5_R_ and H-5_L_ of each two form), 4.50 (1H, *J* = 7.8 Hz, H-1_L_ of β-β form), 5.01~5.34 (complicated peaks, H-2_R_ and 2_L_, H-4_R_ and H-4_L_, H-6a_R_ and H-6a_L_ of each two form), 5.35 (0.5H, d, *J* = 3.6 Hz, H-1_L_ of α-β form), 5.46~5.85 (each t, *J* = 9.6 Hz, H-3_R_ and H-3_L_ of each two from), 6.18 (0.5H, t, *J* = 8.4 Hz, H-1_R_ of α-β form), 6.19 (1H, s, val H_B_ of β-β form), 6.21 (0.5H, s, val H_B_ of α-β form), 6.22 (1H, t, *J* = 8.4 Hz, H-1_R_ of β-β form), 6.51 (1H, s, HHDP of β-β form), 6.53 (0.5H, s, HHDP of α-β form), 6.60 (0.5H, s, val H_A_ of α-β form), 6.53 (0.5H, s, HHDP of α-β), 6.60 (0.5H, s, val H_A_ of α-β form), 6.63 (1H, s, val H_A_ of β-β form), 6.65 (0.5, s, HHDP of α-β form), 6.68 (1H, s, HHDP of β-β form), 6.85 (1H, s, galloyl H-2, 6 of β-β form), 6.92 (0.5H, s, galloyl H-2, 6 of α-β form), 7.00–7.14 (each s, 3 × galloyl H-2, 6; val H_C_ of each two from). 

^13^C-NMR (150 MHz, Aceotone-*d*_6_ + D_2_O): *δ* 62.5–63.0 (C-6_L_ and C-6_R_ of each two form), 66.1 (C-5_L_ of α-β form), 70.0–71.0 (C-4_L_ and C-4_R_ of each two form; C-3_L_ of α-β form, C-5_R_ of β-β form, C-5_L_ of β-β form), 71.9–73.2 (C-2_L_ and C-2_R_ of each two form; C-3_R_ of α-β form, C-3_R_ of β-β form, C-3_L_ of β-β form), 90.1 (C-1_L_ of α-β form), 92.6–92.8 (C-1_R_ of α-β form, C-1_R_ of β-β form, C-1_L_ of β-β form), 104.0 (val C-3′), 106.9(–)107.2 (val C-3, HHDP C-3, HHDP C-3′), 109.0–109.9 (gal C-2 gal C-6; val C-6”), 113.0–115.2 (HHDP C-1, HHDP C-1′; val C-1, val C-1′, valC-1”), 118.7–119.7 (gal C-1), 124.1–125.7 (HHDP C-2, HHDP C-2′; val C-2, val C-2′), 135.0–136.6 (HHDP-5, HHDP-5′; val C-5, val C-5′, val-2”), 138.3–138.6 (gal C-4), 138.9–139.6 (val C-3”, val C-4”), 142.2–142.5 (val C-5”), 143.5–145.1 (gal C-3, gal C-5; HHDP C-4, HHDP C-4′, HHDP C-6, HHDP C-6′; val C-4, val C-6, val C-6′), 145.7–146.7 (val C-4′), 163.9–167.7 (gal C-7; HHDP C-7, HHDP C-7′; val C-7, valC-7”).

#### 4.1.2. Cornusiin A (2)

The product was an amorphous yellowish white powder. Structural data were as follows:

LRFAB-MS *m*/*z*: 1569 [M − H]^(-)^; CD (MeOH): [*θ*] 221 20.85, [*θ*] 259 -2.96, [*θ*] 281 9.22

^1^H-NMR (600 MHz, Acetone-*d_6_* + D2O): *δ* 3.76–3.96 (each br d, *J* = 13.2 Hz, H-6b_L_ and H-6b_R_ of each four form), 4.17–4.24, 4.42–4.45 (each br dd, *J* = 6.6, 10.2 Hz, H-5_R_ of α-β form, H-5_L_ of β-α form, H-5_R_ of β-β form, H-5_L_ of β-β form), 4.48 (d, *J* = 7.8, H-1_L_ of β-β form), 4.52 (d, *J* = 7.8 H-1_L_ of β-α form), 4.60–4.65, 4.73–4.82 (each br dd, *J* = 6.6, 10.2 Hz, H-5_R_ of α-α form, H-5_L_ of α-α of form, H-5_L_ of α-β form, H-5_R_ of β-α form), 5.03–5.11 (complicated peaks, H-2_R_ of α-α form, H-2_L_ of α-α form, H-2_L_ of α-β form, H-2_R_ of β-α form; H-4_L_ and H-4_R_ of each four form), 5.20–5.27, 5.43–5.52 (complicated peaks, H-6a_L_ and H-6a_R_ of each four form; H-3_R_ of α-β form, H-3_L_ of β-α form, H-3_R_ of β-β form, H-3_L_ of β-β form), 5.15 (each d, *J* = 8.4 Hz, H-1_R_ of α-β form, H-1_R_ of β-β form), 5.13–5.18 (each dd, *J* = 8.4, 9.6 Hz, H-2_R_ of α-β form, H-2_L_ of β-α form, H-2_R_ of β-β form, H-2_L_ of β-β form), 5.37–5.39 (each d, *J* = 3.6 Hz, H-1_L_ of α-β, H-1_L_ of α-α form), 5.53 (d, *J* = 3.6 Hz, H-1_R_ of α-α form), 5.56 (d, *J* = 3.6 Hz, H-1_R_ of β-α form), 5.69~5.85 (each t, *J* = 9.6 Hz, H-3_R_ of α-α form, H-3_L_ of α-α form, H-3_L_ of α-β form, H-3_R_ of β-α form), 6.21, 6.22, 6.23, 6.26 (each s, val H_B_), 6.51, 6.52, 6.54, 6.54 (each s, HHDP), 6.64, 6.65, 6.66, 6.66 (each s, val H_A_), 6.69, 6.69, 6.69, 6.71 (each s, HHDP), 6.84, 6.90, 6.92, 6.95 (each s, galloyl 2H), 7.03–7.13 (each s, 2 × galloyl 2H, val H_C_)

^13^C-NMR (150 MHz, Acetone-*d*_6_ + D_2_O): δ 62.7–63.2 (C-6_R_ and C-6_L_ of each four form), 66.1–66.2 (C-5_R_ of α-α form, C-5_L_ of α-α form, C-5_L_ of α-β form, C-5_R_ of β-α form), 72.5–73.5 (C-3_R_ of α-β form, C-3_L_ of β-α form, C-3_R_ of β-β form, C-3_L_ of β-β form; C-2_R_ of α-β form, C-2_L_ of β-α form, C-2_R_ of β-β form, C-2_L_ of β-β form), 70.2–71.1 (C-4_L_ of each four form, C-4_R_ of each four form; C-3_R_ of α-α form, C-3_L_ of α-α form, C-3_L_ of α-β form, C-3_R_ of β-α form; C-5_R_ of α-β form, C-5_L_ of β-α form, C-5_R_ of β-β form, C-5_L_ of β-β form), 72.1–72.3 (C-2_R_ of α-α form, C-2_L_ of α-α form, C-2_L_ of α-β form, C-2_R_ of β-α form), 90.1–90.3 (C-1_R_ of α-α form, C-1_L_ of α-α form, C-1_L_ of α-β form, C-1_R_ of β-α form), 95.3–95.4 (C-1_R_ of α-β form, C-1_L_ of β-α form, C-1_R_ of β-β form, C-1_L_ of β-β form), 104.2–104.4 (val C-3′), 106.8–107.0 (val C-3, HHDP C-3, HHDP C-3′), 109.1–109.4 (gal C-2 gal C-6; val C-6″), 113.2–115.3 (HHDP C-1, HHDP C-1′; val C-1, val C-1″), 116.5~116.7 (val C-1′), 119.6~120.1 (gal C-1), 124.6–125.6 (HHDP C-2, HHDP C-2′; val C-2, val C-2′), 135.1–136.8 (HHDP-5, HHDP-5′; val C-5, val C-5′, val-2″), 138.2–138.4 (gal C-4), 139.3–139.8 (val C-3″, val C-4″), 142.3–142.5 (val C-5″), 143.6–145.1 (gal C-3, gal C-5; HHDP C-4, HHDP C-4′, HHDP C-6, HHDP C-6′; val C-4, val C-6, val C-6′), 145.7–146.5 (val C-4′), 163.8–166.2 (gal C-7), 166.8–168.1 (HHDP C-7, HHDP C-7′; val C-7, val C-7′ val C-7″).

### 4.2. Cell Culture

Murine macrophage RAW 264.7 cells were purchased from the Korean Cell Line Bank (Seoul, Korea). These cells were grown at 37 °C in a humidified atmosphere (5% CO_2_) in Dulbecco′s Modified Eagle′s Medium (DMEM; Sigma-Aldrich, St. Louis, MO, USA), supplemented with 10% fetal bovine serum (FBS), 100 IU/mL penicillin G, and 100 mg/mL streptomycin (Gibco BRL, Grand Island, NY, USA). The cells were used after counting, using a hemocytometer. Human monocytic leukemia THP-1 cells purchased from the Korean Cell Line Bank were grown at 37 °C in a humidified atmosphere (5% CO_2_), in an RPMI 1640 medium (Sigma-Aldrich) supplemented with 10% FBS and 100 IU/mL penicillin G (Thermo Fisher Scientific Korea Ltd., Seoul, Korea). These were used after counting, using a hemocytometer.

### 4.3. Measurement of Cytokines Production

The concentrations of cytokines (IL-1β, IL-6, IL-8, and TNF-α in the culture supernatants were measured by an enzyme-linked immunosorbent assay (ELISA; eBioscience, San Diego, CA, USA). The cytokine concentrations were quantified by measuring the absorbance of the samples at 405 nm with an ELISA reader (TECAN). The levels of cytokines produced were calculated using a standard calibration curve. After the THP-1 cells were exposed to LPS, the cytokine levels were measured to determine the inhibitory effect of the compounds (**1** and **2**), which were used at a concentration 25 μM.

### 4.4. Apoptosis Activity

Cells (2 × 105 cells/well) were cultured in a 6-well plate for 24 h, and then, treated with compounds **1** or **2** (50 μM). The cells were then harvested and washed once using ice-cold PBS, after which they were resuspended in a binding buffer, and stained with 5 μL Annexin V-FITC and 5 μL of PI, for 15 min, in the dark, at room temperature. The fluorescence was analyzed by flow cytometry (BD-LSR II, San Jose, CA, USA) using the “Cell Quest 2.0” software. At least 10,000 events were recorded and represented as dot plots. The percentage of cells in the upper left (necrotic cells), upper right (late apoptotic cells), lower right (early apoptotic cells), and lower left (viable cells) portion of the histogram was calculated for comparison.

### 4.5. Measurement of Inhibitory 5α-Reductase Activity

The 5 α-reductase enzyme from rat liver microsomes was incubated with 400 μL of phosphate buffer (pH 6.5) in the Intact group, and 200 μL of phosphate buffer in the normal group (negative control). Testosterone was used as a substrate for 5α-reductase at a volume of 50 μL (100 μg/mL). The other samples were incubated with 200 μL (1 mM) of finasteride, which was considered the positive control group (finasteride), or 200 μL (1 mM) of compounds (**1** and **2**). Finally, 20 μL of NADPH (0.8 mg/mL) was added. Microsomal enzymes (5α-reductase) isolated from rat livers were added to the samples from all groups, except for the intact group. The reaction was terminated by adding 0.5 mL of dichloromethane for all group. The amount of testosterone in the samples was measured by HPLC. The injection volume was 20 μL and elution was performed at a flow rate of 1 mL/min, using a binary gradient of water (A) and acetonitrile (ACN) (B). The quantification wavelength of these chromatograms was set at 242 nm, which was optimized for testosterone. The data were integrated using the Empower software system (Waters, Coastal, CT, USA).

### 4.6. Macrophage Differentiation and Stimulation

The mature macrophage-like state was induced by treating the THP-1 monocytes (105 cells/mL) for 48 h with 10 nmol of 12-*O*-tetradecanoylphorbol-13-acetate (TPA; Sigma-Aldrich), in 24-well cell culture plates, with 1 mL of cell suspension in each well. Differentiated plastic-adherent cells were washed once with phosphate-buffered saline (PBS) and cultured in a fresh RPMI 1640 medium (Sigma-Aldrich) supplemented with 10% FBS and 100 IU/mL penicillin G (Gibco BRL). Differentiated THP-1 cells were treated with the test samples and 0.1 μg/mL of LPS (Sigma-Aldrich) for 1 h at 37 °C, in a humidified atmosphere (5% CO_2_). After further incubation for 24 h, the supernatants were transferred to Eppendorf tubes, for cytokine assays.

### 4.7. Preparation of Liver Microsomes

Liver microsomes were prepared from the liver of male rats. Two mature Sprague–Dawley male rats were euthanized, and their livers were removed and minced in a beaker, with a pair of scissors. The minced tissue was homogenized in three volumes of medium A (0.32 M sucrose, 1 mM dithiothreitol, and 20 mM sodium phosphate, pH 6.5) and the homogenate was centrifuged at 10,000× *g* for 10 min. The resulting pellet was washed with two volumes of medium A. The combined supernatant from the two centrifugations was suspended in 4 mL of medium A, and the dispersion of the microsomes was achieved using a syringe with 18 G, 23 G, and 25 G needles, in succession. The microsome suspension was divided into aliquots and stored at −80 °C. The microsomes were diluted with the medium just before use.

### 4.8. Statistical Analyses

The results were analyzed by one-way analysis of variance (ANOVA), followed by the Student–Newman–Keuls (S–N–K) test and one to one confrontation test, to determine the *t*-value and *p*-value, using the Statistical Package for the Social Sciences (SPSS) software pack (IBM, Armonk, NY, USA).

## 5. Conclusions

The two ellagitannins [camptothin B (**1**), cornusiin A (**2**)] were isolated from *Cornus alba* (CA). The two ellagitannins (**1** and **2**) showed potent inhibitions of cytokine (IL-6, IL-8, IL-1β and TNF-α) production, as well as anti-proliferative activities against the RWPE-1 cell. Moreover, they were also shown to induce apoptosis in the cell and inhibit the 5α-reductase activity, as compared to finasteride. Hence, the ellagitannins (**1** and **2**) isolated from CA may be good candidates for the treatment of BPH.

## Figures and Tables

**Figure 1 molecules-26-03446-f001:**
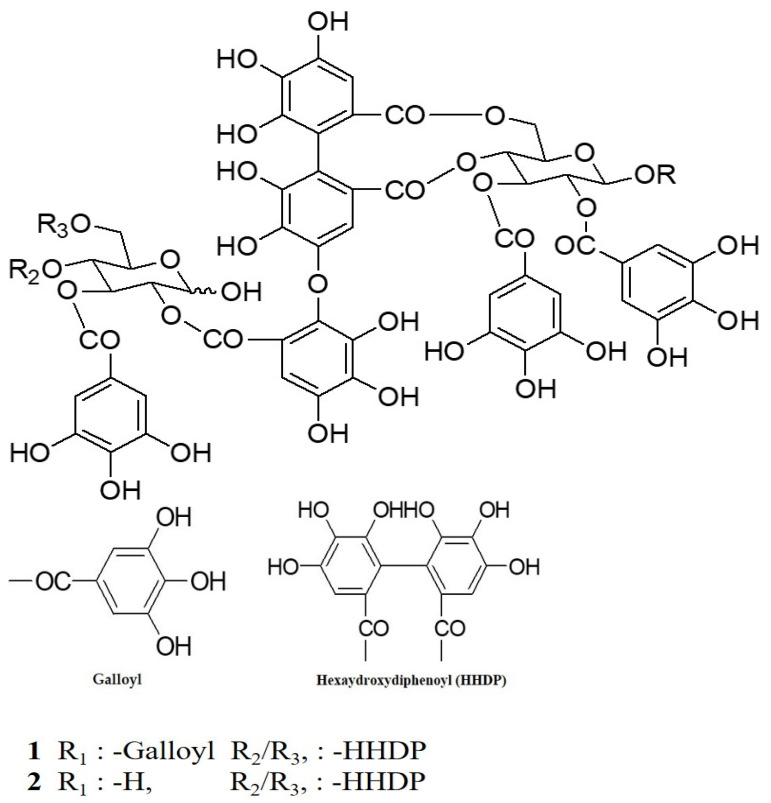
The structures of compounds **1** and **2** isolated from *Cornus alba*.

**Figure 2 molecules-26-03446-f002:**
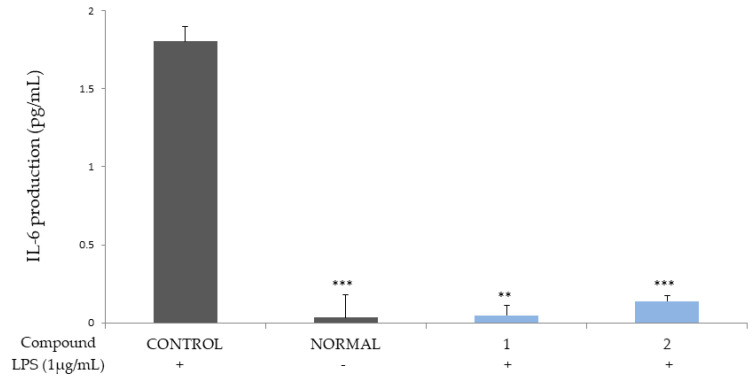
Inhibitory activities of compounds **1** and **2** on IL-6 production in the THP-1 macrophage cell. Control, compounds **1 and 2** were added LPS (1 μg/mL). Normal was not added LPS. Values are expressed as the means ± standard deviation (SDs) from triplicate measurements. * *p* < 0.05, ** *p* < 0.01, and *** *p* < 0.001.

**Figure 3 molecules-26-03446-f003:**
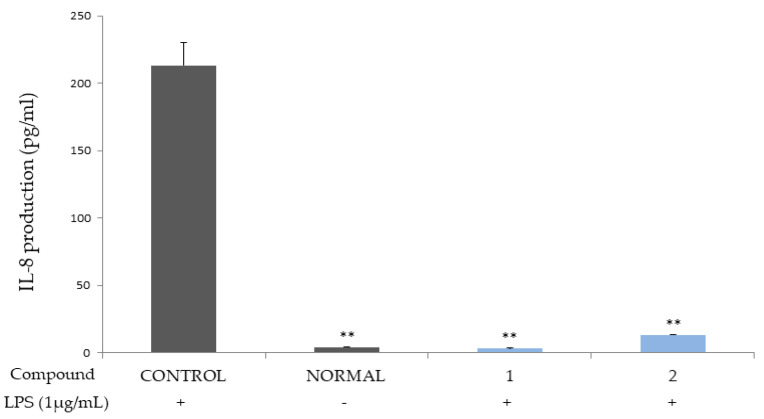
Inhibitory activities of compounds **1** and **2** on IL-8 production in THP-1 macrophage cell. Control, compounds **1 and 2** were added LPS (1 μg/mL). Normal was not added LPS. Values are expressed as the means ± standard deviations (SDs) from triplicate measurements. * *p* < 0.05, and ** *p* < 0.01.

**Figure 4 molecules-26-03446-f004:**
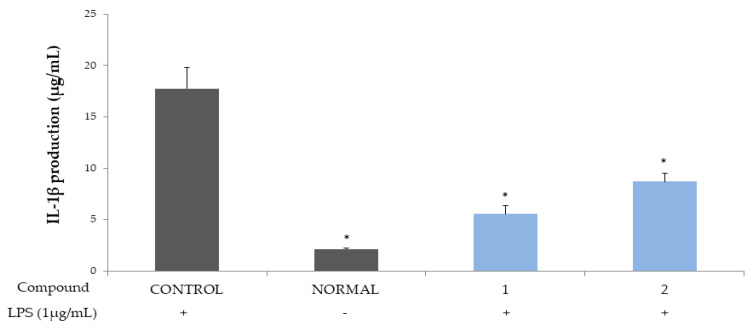
Inhibitory activities of compounds **1** and **2** on IL-1β production in the THP-1 macrophage cell. Control, compounds **1 and 2** were added LPS (1 μg/mL). Normal was not added LPS. Values are expressed as the mean ± standard deviations (SDs) from triplicate measurements. * *p* < 0.05.

**Figure 5 molecules-26-03446-f005:**
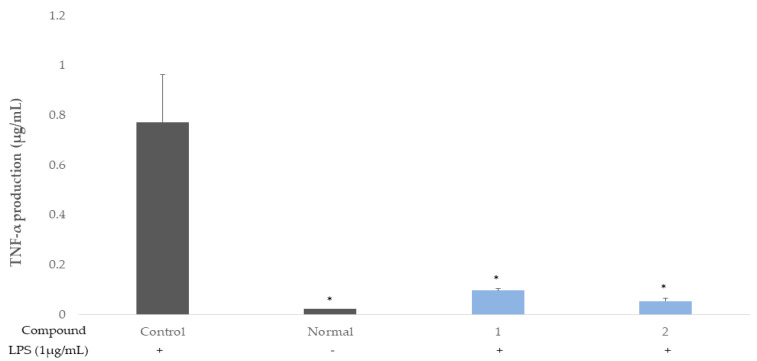
Inhibitory activities of **1** and **2** on TNF- α production in the THP-1 macrophage cell. Control, compounds **1 and 2** were added LPS (1 μg/mL). Normal was not added LPS. Values are expressed as means ± standard deviations (SDs) from triplicate measurements. * *p* < 0.05.

**Figure 6 molecules-26-03446-f006:**
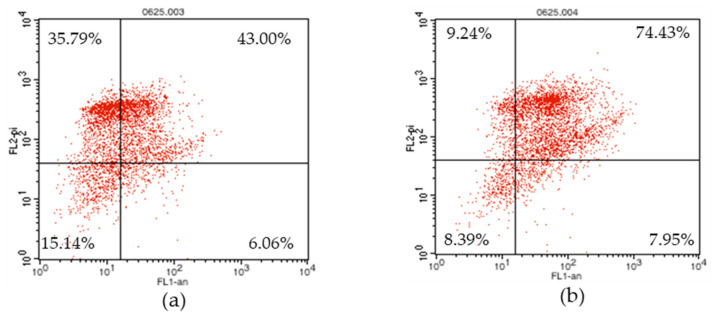
Flow cytometry analysis of apoptosis via annexin V/PI dual staining in the RWPE-1 cell. (**a**) Flow cytometry analysis of apoptosis treated with compound **1**. (**b**) Flow cytometry analysis of apoptosis treated with compound **2**.

**Figure 7 molecules-26-03446-f007:**
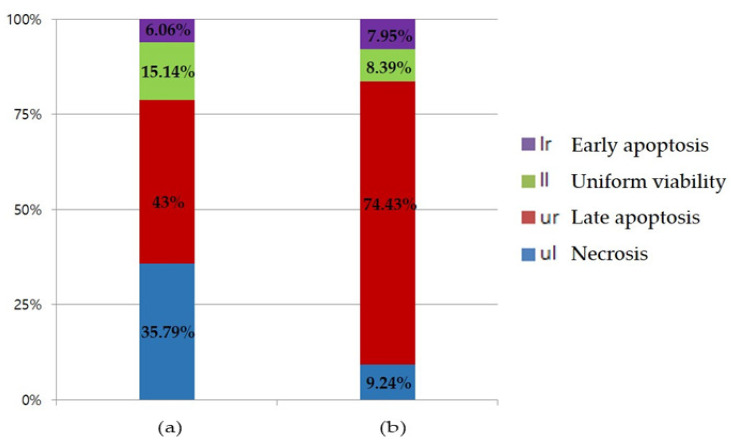
Flow cytometry (FACS) analysis of compounds **1** and **2**. (**a**) represents compound **1** and (**b**) represents compound **2**.

**Figure 8 molecules-26-03446-f008:**
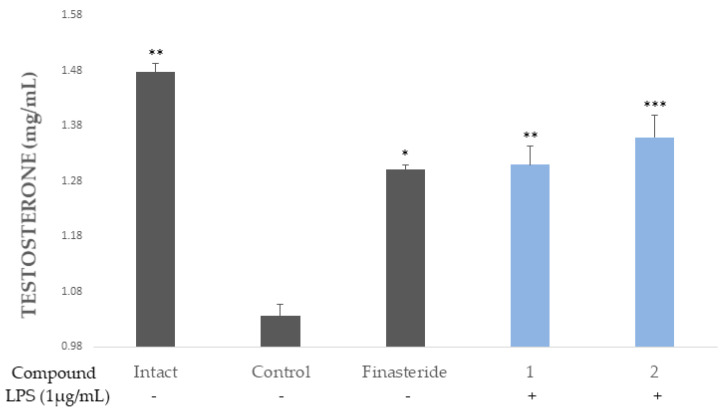
Compounds **1** and **2** inhibit 5α-reductase activity. Values are expressed as the means ± standard deviations (SDs) from the triplicate measurements. * *p* < 0.05, ** *p* < 0.01, *** *p* < 0.001.

## Data Availability

The data are available in this article. Please inquiries about the data of the thesis can be made via e-mail to mwlee@cau.ac.kr.

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
