# Peer review of "Inhibitory Activities of Dimeric Ellagitannins Isolated from Cornus alba on Benign Prostatic Hypertrophy"

_molecules, 2021, doi:10.3390/molecules26113446_

Round 1

Reviewer 1 Report

Interesting paper but especially the quality of its presentation must be improved. 

Abstract should follow the structure of the manuscript. Please add a short first phrase regarding the background presenting the topic.

The sequence in the manuscript of two consecutive references must be made as [a,b], not [a-b] (ie L38, references [6,7]

Too much empty space between the paragraphs of the same section. Please remove it.

1. Introduction section needs to be developed a little more.

L50. Please reshape the sentence beginning with "And decocted...". It sounds strange.

Information presented in Table 1 must be reshaped as text, and the Table should be removed (there are just 2 numerical data, no needed for a Table)

4.1. Phytochemicals

Even the authors have mentioned that "The plant materials and compound isolation procedures have been described previously [16].", I consider that they must to add information describing the chemical reagents (adding also their CAS), apparatus (producer, town, country of manufacturer), plant material and isolation procedure.

Conclusion section is missing. I know that it is not a mandatory one (according to the Molecules Instructions for authors), but it always highlights the main/special findings of a study and its novelty character. Maybe the authors will reconsider to insert this section in their manuscript. L173-178 can be easily moved to this new section.

Adding information in the sections I mentioned will also contribute to the length of the manuscript. A paper of only 9 effective pages (and if the empty spaces will be removed, the manuscript will result under 9 pages) seems too short for a journal like Molecules. Indeed, the data is concentrated/focused on the topic, but in my opinion, it needs to be presented in a broader way.

Author Response

Interesting paper but especially the quality of its presentation must be improved. 

Abstract should follow the structure of the manuscript. Please add a short first phrase regarding the background presenting the topic.

- We added a sentence about the background explaining the topic at the first phrase of the abstract. The last sentence may also indicate the final outcome of this study.

Abstract: Benign prostatic hypertrophy (BPH) is intractable chronic inflammatory disease. We studied the efficacy of two ellagitannins, namely camptothin B (1) and cornusiin A (2) and Cornus alba (CA) extract for the treatment of benign prostatic hypertrophy (BPH), which is a common health issue in older men. The ellagitannins (1 and 2) were evaluated on its inhibitory activities of the enzyme 5α-reductase and tumor necrosis factor (TNF)-α, as well as interleukin (IL)-1β, IL-6, and IL-8 production and its anti-proliferation and inducing apoptosis in, prostate cells showing hypertrophy (RWPE-1 cells). In inhibition of 5α-reductase, the ellagitannins (1 and 2) showed potent effects compared with the positive control, finasteride. In the case of IL-1β, IL-6, IL-8 and TNF-α, 1 and 2 showed good inhibitory effects compared with the control group treated with LPS. The ellagitannins (1 and 2) were also shown to inhibit proliferation of, and induce apoptosis in, RWPE-1 cells. These results suggest that the ellagitannins (1 and 2), may be good candidates for the treatment of BPH.

The sequence in the manuscript of two consecutive references must be made as [a,b], not [a-b] (ie L38, references [6,7]

- The order of the two consecutive reference manuscripts has been modified as requested.

Studies have reported that prostatic growth begins at age 30, with and BPH affecting approximately 50% of men aged 50 and 70% of men older than 80 [4,5]. The cause of BPH is unclear, however, studies have suggested that the common factors that might to induce the development of BPH are hormonal alterations, ageing and inflammation [6,7].

Too much empty space between the paragraphs of the same section. Please remove it.

- As requested, we have reduced the spaces between the same sections as much as possible.

  1. Introduction section needs to be developed a little more.

L50. Please reshape the sentence beginning with "And decocted...". It sounds strange.

- We revised it as below

Cornus alba (CA), is a species of Cornaceae, native to Siberia, northern China and Korea. It is a deciduous shrub that can grow to 3 m high like a small tree [11]. The barks and leaves of CA were used as an folk herbal medicine for the treatment of inflammation and hemolysis [12]. And CA bark is used for the hemoptysis of pulmonary tuberculosis, and the root is used for fever and cold [13]. Cornus species are known to have antioxidant, anti-inflammatory and anti-cancer biological activities also [14,15].

Information presented in Table 1 must be reshaped as text, and the Table should be removed (there are just 2 numerical data, no needed for a Table)

- We revised it as your comment as below

2.3. Anti-proliferative effect on BPH cell line

The anti-proliferative effect of the compounds on the BPH cell line (RWPE-1 cell) was measured via an MTT assay after the cell was treated with the compounds at concentrations between 6.25-100μM for 24 h. The ellagitannins (1 and 2) were shown to have anti-proliferative activities with half maximal inhibitory concentrations (IC50) of 3.38 ± 0.71 and 1.54 ± 0.21, respectively.

4.1. Phytochemicals

Even the authors have mentioned that "The plant materials and compound isolation procedures have been described previously [16].", I consider that they must to add information describing the chemical reagents (adding also their CAS), apparatus (producer, town, country of manufacturer), plant material and isolation procedure.

- As you mentioned, information describing the plant's (producer, village, country) has been added, and procedures have also been added as below

4.1. Compounds

CA (5.7 kg) was collected at the Korea National Arboretum (Pocheon, Korea). The CA samples (5.7 kg) were pulverized and extracted with 80% acetone at room temperature to obtain the CA extract (463 g). Then, the extracted and its subfraction (356 g) were separated via repeated column chromatography. Next, the samples were dissolved in water and filtered through a Celite 545 (Duksan Pure Chemical, Ansan, Korea) filter, after which 356 g of a water-soluble fraction was obtained together with 89 g of water-insoluble residue. Only 243 g of the water-soluble fraction was loaded onto a Sephadex LH-20 column (15 x 100 cm) equilibrated with water. After the column was eluted with a water-methanol gradient system and washed with 60% acetone, 14 fractions were obtained. Fraction 13 (31.6 g) was loaded onto MCI CHP 20P column (5 × 60 cm) with a water -methanol gradient system, and five subfractions were obtained. Afterwards, fraction 13-2 was loaded onto a Daisogel column (3 × 50 cm), with a water 20 % methanol gradient in a Medium pressure Liquid Chromatography (MPLC) system (5 mL/min, 280 nm) and was further separated by column chromatography on the Sephadex LH-20 column (10 × 80 cm) using a water-methanol 60% acetone gradient system. As a result, cornusiin A (2, 1.2 g) was obtained. Fraction 13-3 was loaded onto a Sephadex LH-20 column (2.5 × 50 cm) with a water-methanol 60% acetone gradient system and further separated by column chromatography on the MCI CHP20P column (5 × 60 cm) using a water-methanol gradient system. This resulted in three additional sub-fractions. Fraction 13-3-1 was then separated by column chromatography using a Toyopearl HW-40 column (2.5 × 50 cm) with a 70% methanol – 70% acetone (10 : 0 → 7:3) gradient, which yielded camptothin B (1, 815 mg).

Camptothin B (1)

The product was an amorphous brown powder. Structural data are as follows:

LRFAB-MS m/z: 1721 [M-H]-

CD (MeOH): [θ]223 16.13, [θ]259 3.18, [θ]280 6.36

1H-NMR (600 MHz, Acetone-d6+D2O) : δ 3.73-3.96 (each br d, J=13.2, H-6b of each two form), 4.14, 4.48, 4.59-4.65 (br dd, J=6.0, 9.6 Hz, H-5R and H-5L of each two form), 4.50 (1H, J=7.8 Hz, H-1L of β-β form), 5.01-5.34 (complicated peaks, H-2R and 2L, H-4R and H-4L, H-6aR and H-6aL of each two form), 5.35 (0.5H, d, J=3.6 Hz, H-1L of α-β form), 5.46-5.85 (each t, J=9.6 Hz, H-3R and H-3L of each two from), 6.18 (0.5H, t, J=8.4 Hz, H-1R of α-β form), 6.19 ( 1H, s, val HB of β-β form), 6.21 (0.5H, s, val HB of α-β form), 6.22 (1H, t, J=8.4 Hz, H-1R of β-β form), 6.51 (1H, s, HHDP of β-β form), 6.53 (0.5H, s, HHDP of α-β form), 6.60 (0.5H, s, val HA of α-β form), 6.53 (0.5H, s, HHDP of α-β), 6.60 (0.5H, s, val HA of α-β form), 6.63 (1H, s, val HA of β-β form), 6.65 (0.5, s, HHDP of α-β form),

6.68 (1H, s, HHDP of β-β form), 6.85 (1H, s, galloyl H-2, 6 of β-β form), 6.92 (0.5H, s, galloyl H-2, 6 of α-β form), 7.00-7.14 (each s, 3 × galloyl H-2, 6 ; val HC of each two from)

13C-NMR (150 MHz, Aceotone-d6 + D2O) : δ 62.5-63.0 (C-6L and C-6R of each two form), 66.1 (C-5L of α-β form), 70.0-71.0 (C-4L and C-4R of each two form; C-3L of α-β form, C-5R of β-β form, C-5L of β-β form), 71.9-73.2 (C-2L and C-2R of each two form; C-3R of α-β form, C-3R of β-β form, C-3L of β-β form), 90.1 (C-1L of α-β form), 92.6-92.8 (C-1R of α-β form, C-1R of β-β form, C-1L of β-β form), 104.0 (val C-3’), 106.9-107.2 (val C-3, HHDP C-3, HHDP C-3’), 109.0-109.9 (gal C-2 gal C-6; val C-6”), 113.0-115.2 (HHDP C-1, HHDP C-1’; val C-1, val C-1’, valC-1”), 118.7-119.7 (gal C-1), 124.1-125.7 (HHDP C-2, HHDP C-2’; val C-2, val C-2’), 135.0-136.6 (HHDP-5, HHDP-5’; val C-5, val C-5’, val-2”), 138.3-138.6 (gal C-4), 138.9-139.6 (val C-3”, val C-4”), 142.2-142.5 (val C-5”), 143.5- 145.1 (gal C-3, gal C-5; HHDP C-4, HHDP C-4’, HHDP C-6, HHDP C-6’; val C-4, val C-6, val C-6’), 145.7-146.7 (val C-4’), 163.9-167.7 (gal C-7; HHDP C-7, HHDP C-7’; val C-7, valC-7”)

Cornusiin A (2)

The product was an amorphous yellowish white powder. Structural data are as follows:

LRFAB-MS m/z: 1569 [M-H]-

CD (MeOH): [θ]221 20.85, [θ]259 -2.96, [θ]281 9.22

1H-NMR (600 MHz, Acetone-d6 +D2O): δ 3.76-3.96 (each br d, J = 13.2 Hz, H-6bL and H-6bR of each four form), 4.17-4.24, 4.42-4.45 (each br dd, J = 6.6, 10.2 Hz, H-5R of α-β form, H-5L of β-α form, H-5R of β-β form, H-5L of β-β form), 4.48 (d, J = 7.8, H-1L of β-β form), 4.52 (d, J = 7.8 H-1L of β-α form), 4.60-4.65, 4.73-4.82 (each br dd, J = 6.6, 10.2 Hz, H-5R of α-α form, H-5L of α-α of form, H-5L of α-β form, H-5R of β-α form), 5.03-5.11 (complicated peaks, H-2R of α-α form, H-2L of α-α form, H-2L of α-β form, H-2R of β-α form; H-4L and H-4R of each four form), 5.20-5.27, 5.43-5.52 (complicated peaks, H-6aL and H-6aR of each four form; H-3R of α-β form, H-3L of β-α form, H-3R of β-β form, H-3L of β-β form), 5.15 (each d, J = 8.4 Hz, H-1R of α-β form, H-1R of β-β form), 5.13-5.18 (each dd, J = 8.4, 9.6 Hz, H-2R of α-β form, H-2L of β-α form, H-2R of β-β form, H-2L of β-β form), 5.37-5.39 (each d, J = 3.6 Hz, H-1L of α-β, H-1L of α-α form), 5.53 (d, J = 3.6 Hz, H-1R of α-α form), 5.56 (d, J = 3.6 Hz, H-1R of β-α form), 5.69-5.85 (each t, J = 9.6 Hz, H-3R of α-α form, H-3L of α-α form, H-3L of α-β form, H-3R of β-α form), 6.21, 6.22, 6.23, 6.26 (each s, val HB), 6.51, 6.52, 6.54, 6.54 (each s, HHDP), 6.64, 6.65, 6.66, 6.66 (each s, val HA), 6.69, 6.69, 6.69, 6.71 (each s, HHDP), 6.84, 6.90, 6.92, 6.95 (each s, galloyl 2H), 7.03-7.13 (each s, 2 × galloyl 2H, val HC)

13C-NMR (150 MHz, Acetone-d6 +D2O): δ 62.7-63.2 (C-6R and C-6L of each four form), 66.1-66.2 (C-5R of α-α form, C-5L of α-α form, C-5L of α-β form, C-5R of β-α form), , 72.5-73.5 (C-3R of α-β form, C-3L of β-α form, C-3R of β-β form, C-3L of β-β form; C-2R of α-β form, C-2L of β-α form, C-2R of β-β form, C-2L of β-β form), 70.2-71.1 (C-4L of each four form, C-4R of each four form; C-3R of α-α form, C-3L of α-α form, C-3L of α-β form, C-3R of β-α form; C-5R of α-β form, C-5L of β-α form, C-5R of β-β form, C-5L of β-β form), 72.1-72.3 (C-2R of α-α form, C-2L of α-α form, C-2L of α-β form, C-2R of β-α form), 90.1-90.3 (C-1R of α-α form, C-1L of α-α form, C-1L of α-β form, C-1R of β-α form), 95.3-95.4 (C-1R of α-β form, C-1L of β-α form, C-1R of β-β form, C-1L of β-β form), 104.2-104.4 (val C-3’), 106.8-107.0 (val C-3, HHDP C-3, HHDP C-3’), 109.1-109.4 (gal C-2 gal C-6; val C-6’’), 113.2-115.3 (HHDP C-1, HHDP C-1’; val C-1, val C-1’’), 116.5-116.7 (val C-1’), 119.6-120.1 (gal C-1), 124.6-125.6 (HHDP C-2, HHDP C-2’; val C-2, val C-2’), 135.1-136.8 (HHDP-5, HHDP-5’; val C-5, val C-5’, val-2’’), 138.2-138.4 (gal C-4), 139.3-139.8 (val C-3’’, val C-4’’), 142.3-142.5 (val C-5’’), 143.6-145.1 (gal C-3, gal C-5; HHDP C-4, HHDP C-4’, HHDP C-6, HHDP C-6’; val C-4, val C-6, val C-6’), 145.7-146.5 (val C-4’), 163.8-166.2 (gal C-7), 166.8-168.1 (HHDP C-7, HHDP C-7’; val C-7, val C-7’ val C-7’’).

Conclusion section is missing. I know that it is not a mandatory one (according to the Molecules Instructions for authors), but it always highlights the main/special findings of a study and its novelty character. Maybe the authors will reconsider to insert this section in their manuscript. L173-178 can be easily moved to this new section.

( L173-178 ) Conclusively, the two ellagitannins, camptothin B (1) and cornusiin A (2) and CA extract showed potent inhibitions of cytokines (IL-6, IL-8, IL-1β and TNF-α) production, as well as anti-proliferative activities against BPH cells. Moreover, they were also shown to induce apoptosis in RWPE-1 cell and the ellagitannins (1 and 2) showed potent 5 α-reductase inhibitory activity. Hence, the ellagitannins (1 and 2) and CA extracts, may be good candidates for the treatment of BPH.

- We have completed an additional conclusion section. By adding a conclusion section, we can once again highlight the main/special findings of the study.

  1. Conclusions

The two ellagitannins [camptothin B (1), cornusiin A (2)] were isolated from Cornus alba (CA). CA extract and two ellagitannins (1 and 2) showed potent inhibitions of cytokine (IL-6, IL-8, IL-1β and TNF-α) production, as well as anti-proliferative activities against  RWPE-1 cell. Moreover, they were also shown to induce apoptosis in these cells and inhibited 5α-reductase activity compared with finateride. Hence, the ellagitannins (1 and 2) isolated from CA, may be good candidates for the treatment of BPH..

Adding information in the sections I mentioned will also contribute to the length of the manuscript. A paper of only 9 effective pages (and if the empty spaces will be removed, the manuscript will result under 9 pages) seems too short for a journal like Molecules. Indeed, the data is concentrated/focused on the topic, but in my opinion, it needs to be presented in a broader way.

- According to your valuable comment we can make the pages from 9 to 13 including phytochemical features.

Reviewer 2 Report

The quality of the article is strongly increased following the current modifications. Although numerous in vitro experiments remain to be done before a good conclusion about the possible use in humans I estimate that your results can be published.

Author Response

The quality of the article is strongly increased following the current modifications. Although numerous in vitro experiments remain to be done before a good conclusion about the possible use in humans I estimate that your results can be published.

As you said, the thesis was revised by supplementing a little more.

  1. Conclusions

The two ellagitannins [camptothin B (1), cornusiin A (2)] were isolated from Cornus alba (CA). CA extract and two ellagitannins (1 and 2) showed potent inhibitions of cytokine (IL-6, IL-8, IL-1β and TNF-α) production, as well as anti-proliferative activities against  RWPE-1 cell. Moreover, they were also shown to induce apoptosis in these cells and inhibited 5α-reductase activity compared with finateride. Hence, the ellagitannins (1 and 2) isolated from CA, may be good candidates for the treatment of BPH..

Round 2

Reviewer 1 Report

The authors responded to all my requests.

This manuscript is a resubmission of an earlier submission. The following is a list of the peer review reports and author responses from that submission.

Round 1

Reviewer 1 Report

This manuscript describes anti BPH activity of dimeric ellagitannins, containing novel information of IL-6, IL-8, IL-1beta, TNF-alpha production inhibition etc. Therefore this can be published after a minor revision considering the following comments.

Line 36, “surgery.,”; remove comma.

Line 38, “inhibitors,.”; remove comma.

Line 44, “from this plant BPH”; “BPH” not necceary?

Line 47,48 “Among them, two ellagitannins (1,2) showed potent inhibitory activities on the production of cytokines interleukin (IL)-6 and 8 (Figure 2,3).”: This sentence had better be deleted because (1) line 46-51 describes the previous results, and (2) the virtually the same sentence appears in the present results section (line 56-58).

Line 88, “Ours”; “Ours” should be deleted.

Figure 2, 3, 4, and 5: The reviewer suggests to reduce the distance between bars in these bar graphs to save the space.

Reviewer 2 Report

The paper would have potential, but it must be rewritten and completed in a much more elaborated way, better documented with References and respecting the structure of a paper as Molecules journal requests..

Abstract is too short, not respecting to follow, in a short form, the structure of the paper. No numerical data. The part regarding conclusions is before the sentence regarding the results.

The Introduction section is extremely weak, incomplete, poorly documented (only 4! References). At the final of the Introduction section, last sentence should be mentioned in a separate paragraph; also, the aim of the study should present the aspect(s) of novelty or the special aspects that this paper brings into the field.

As the Instructions for Author request (please see https://www.mdpi.com/journal/molecules/instructions), present Results and Discussion section should be 2 separate sections. Discussion part is almost completely missing in this form of the article.

Some figures are unclear/unreadable

In Materials and methods sections, for all reagents/chemicals the producer must be provided.

References should be written in MDPI style, as the Instructions for Author request (link mentioned above).

Reviewer 3 Report

Prevention and treatment of human BPH is a remarkably interesting subject. The global approach (chemical purification and in vitro tests on cells) is acceptable. Surprisingly your previous article ( Molecules. 2016 Jan 23;21(2):137. doi: 10.3390/molecules21020137) is not presented in the introduction part (please indicates the first results and the interst of the new one's) . Results in Figure 1 to 5  must be presented as a dose effect. Please explain why you use a concentration of 25 µM while IC50 is 3 or 1 µM ? Same question for concentrations used for 5 alpha reductase activity? Please add a comment on the results published for oral biodisponibility of this type of compounds for a possible human use